# A Buckling Analysis of Thermoelastic Micro/Nano-Beams Considering the Size-Dependent Effect and Non-Uniform Temperature Distribution

**DOI:** 10.3390/ma16196390

**Published:** 2023-09-25

**Authors:** Xin Ren, Shuanhu Shi

**Affiliations:** 1School of Railway Technology, Lanzhou Jiaotong University, Lanzhou 730070, China; 2School of Mechanical and Electrical Engineering, Lanzhou Jiaotong University, Lanzhou 730070, China; shishuanhu08@163.com

**Keywords:** micro/nano-beams, buckling behavior, thermoelastic coupling, non-Fourier heat conduction, non-uniform temperature distribution

## Abstract

Thermoelastic buckling of micro/nano-beams subjected to non-uniform temperature distribution is investigated in this paper. The mechanical governing equation is derived based on the surface effect and mechanical non-local effect. The non-local heat conduction model is used to predict temperature distribution in micro/nano-beams. Therefore, the obtained analytical solution can be used to analyze the influence of both the mechanical and thermal small scale effects on buckling of thermoelastic micro/nano-beams. In numerical simulations, a critical thickness is proposed to determine the influence region of both mechanical and thermal small scale effects. The influence of a small scale effect on buckling of micro/nano-beams must be considered if beam thickness is less than the critical thickness. In the influence region of a small scale effect, a surface effect has strong influence on the size-dependent buckling behavior, rather than mechanical and thermal non-local effects. Moreover, combined small scale effects, i.e., a surface effect and both mechanical and thermal non-local effects, lead to a larger critical load. Additionally, the influence of other key factors on buckling of the micro/nano-beams is studied in detail. This paper provides theoretical explanation to the buckling behaviors of micro/nano-beams under a non-uniform temperature distribution load.

## 1. Introduction

Micro/nano-beams have been widely used in micro-electro-mechanical systems (MEMS) and nano-electro-mechanical systems (NEMS), such as sensors, actuators, resonators and transistors [1]. In practical applications, micro/nano-beams usually sustain complex thermal environments during a manufacturing and working process [2,3], and are sometimes even simultaneously subjected to axial compressive force and variable temperature, such as basic elements for writing or reading information that are based on the use of contact and heat transfer between an atomic force microscope (AFM) probe and the disk [4], and non-uniform temperature distribution exists in vibrating resonators [5]. Therefore, a buckling analysis of thermoelastic micro/nano-beams has attracted great attentions for years.

Thermoelastic coupling can result in thermal stress when structures are subjected to an external thermal load [6,7,8]. Unfortunately, classical thermoelastic theory fails to predict a small scale effect, i.e., size-dependent thermoelastic behavior of micro/nano-structures. In order to overcome the shortcoming, a new theoretical framework is proposed to describe a mechanical small scale effect [9,10] and non-Fourier heat conduction [11]. For example, Sahmani et al. [12] investigated buckling of a functionally graded nano-beam by introducing the strain gradient into classical third-order shear deformation beam theory. Ke et al. [13] studied buckling of micro-beams considering modified couple stress theory. Free vibration of micro-beams was also discussed by them. Applying the surface elastic theory, thermal buckling of heated nanowire was examined by Wang et al. [14] in the context of Euler–Bernoulli beam theory. From the literature review, it can be concluded that the thermoelastic buckling issues are useful to engineering applications, and which are usually studied based on the uniform temperature distribution, i.e., temperature distribution at every point of a structure is the same. From the point of view of mathematics, uniform temperature distribution assumption in micro/nano-beams can result in a vanished partial derivative, which is a thermal-induced resultant with respect to coordinate variables. Therefore, this simplified assumption on temperature distribution makes a thermoelastic buckling analysis easy.

In fact, structural elements subjected to non-uniform heating widely exist in engineering application. For example, structures subjected to a heat source can result in non-uniform temperature distribution [15], and both non-uniform temperature distribution and axial force exist in vibrating thermoelastic structures due to the interaction of an internal micro-structure [16]. In these cases, thermoelastic buckling of micro/nano-beams based on the hypothesis of uniform temperature distribution is not accurate anymore. Therefore, investigation on micro/nano-beams with non-uniform temperature distribution is required. Lee et al. [17] supposed that temperature was linearly distributed along the length direction for a vibration analysis of a scanning thermal microscope probe. Ebrahimi and Salari [18] studied thermal buckling and free vibration of size-dependent nano-beams assuming that temperature distribution was linearly varied along the thickness. Yu et al. [19] analyzed buckling of beams subjected to non-uniform temperature based on the non-local thermoelastic theory. The non-uniform temperature distribution was conducted with the non-local heat conduction model [20]. Zhang et al. [21] investigated buckling of Bernoulli–Euler beams under non-uniform temperature based on a two-phase non-local integral model. Xu et al. [22] discussed buckling of functionally graded nano-beams under non-uniform temperature using stress-driven non-local elasticity. These literature reviews clearly showed that mechanical behaviors of micro/nano-beams are significantly affected by thermoelastic coupling. On the other hand, from the perspective of physics, at the micro/nano-scale, the mean-free-path of thermal phonons is comparable or even much larger than the material length scale. This phenomenon has great influence on heat conduction behavior [23]. At present, lots of modified Fourier heat conduction models, i.e., the non-Fourier heat conduction models, have been proposed to predict micro/nano-scaled heat conduction behaviors [20,24,25]. However, as mentioned before, extensive review has showed that buckling analyses of thermoelastic micro/nano-beams based on the combined effects of the non-Fourier heat conduction model and mechanical small scale effect are few. Therefore, there is still a lack of fundamental understandings on micro/nano-beams subjected to non-uniform temperature distribution.

In this study, we attempt to investigate buckling of size-dependent micro/nano-beams considering non-uniform temperature distribution. The buckling model is established based on the non-local elastic theory [26] and surface elastic theory [27]. The non-local heat conduction model proposed by Tzou et al. [24] is used to predict temperature distribution in buckling of curved micro/nano-beams. An analytical solution of the buckling issue is obtained. Numerical simulations are presented graphically and discussed to clarify the combined small scale effects on buckling of thermoelastic micro/nano-beams. This paper provides a new basis for the understanding of buckling behavior.

## 2. Formulation of the Problem

As shown in Figure 1, we aim to derive the critical load of thermoelastic micro/nano-beams with length, width and height as *L*, *a* and *h*, respectively. In the theoretical deduction, temperature distribution in the micro/nano-beams is calculated with the non-local heat conduction model, and the influence of the thermal effect on the micro/nano-beams is regarded as an equivalent axial force.

### 2.1. Mechanical Governing Equation

The non-local elastic theory [26] and the surface elastic theory [27] have been widely accepted during micro/nano-structure analyses. The non-local elastic theory is in accordance with atomic theory of lattice dynamics and phonon dispersion. The surface elastic theory [27] describes the distinct environment of atoms between the surface layer and bulk material. A different physical mechanism is described with the two theories above. Therefore, the non-local elastic theory [26] and the surface elastic theory [27] can be simultaneously used to predict mechanical behavior of size-dependent micro/nano-structures [28,29]. Based on the Euler–Bernoulli beam theory, the constitutive equation of micro/nano-beams considering the mechanical non-local effect and surface effect can be expressed as follows [28]:(1)$$M-\xi^{2}\frac{d^{2}M}{dx^{2}}=-{\left(EI\right)}^{*}\frac{d^{2}w}{dx^{2}}$$
where M is the bending moment, ξ is the mechanical non-local parameter and *w* is the transverse displacement. ${\left(EI\right)}^{*}$ is the effective flexural rigidity, which includes the surface bending elasticity and flexural rigidity. The effective flexural rigidity of micro/nano-beams with a rectangular section can be expressed as follows [28,30]:(2)$${\left(EI\right)}^{*}=EI+\bar{E}\bar{I},$$
where $\bar{E}\bar{I}=\frac{1}{2}E_{s}ah^{2}+\frac{1}{6}E_{s}h^{3}$, and $I=\frac{1}{12}ah^{3}$ is the moment of inertia. Parameters *E* and $E^{s}$ are the Young’s modulus and surface Young’s modulus, respectively.

Mechanical equilibrium equations of bending micro/nano-beams can be expressed as follows [28]:(3)$$V=\left(P_{F}-H\right)\frac{dw}{dx},$$
(4)$$V=\frac{dM}{dx},$$
where $P_{F}$ is the axial force, *V* is the shear force and *H* is the constant determined with the residual surface tension τ and cross-section size. For thermoelastic issues, the temperature-induced equivalent thermal load can be regarded as the component of the effective external axial load. In other words, the axial force $P_{F}$ is constituted by two parts: the external axial load P and temperature-induced equivalent thermal load $P_{\theta }$, i.e., $P_{F}=P+P_{\theta }$. These parameters above can be expressed as follows [19,30]:(5)H=2τa,
(6)$$P_{\theta }=\frac{EA}{L\left(1-2\mu \right)}\int_{0}^{L}\alpha_{t}\theta dx,$$
where *A* = *ah* is the cross-sectional area of micro/nano-beams. Parameters $\alpha_{t}$, μ and θ are the thermal expansion coefficient, Poisson’s ratio and the temperature increment between current temperature *T* and reference temperature $T_{0}$, i.e., $\theta =T-T_{0}$.

Eliminating shear force *V* from Equations (3) and (4), and then substituting the result into the second derivative of Equation (1) with respect to coordinate *x*, the governing equation of micro/nano-beams can be obtained:(7)$$\left[{\left(EI\right)}^{*}-\xi^{2}\left(P_{F}+P_{\theta }-H\right)\right]\frac{\partial^{4}w}{\partial x^{4}}+\left(P_{F}+P_{\theta }-H\right)\frac{\partial^{2}w}{\partial x^{2}}=0.$$

The equation presented above takes into account the influences of the surface effect, thermal effect and mechanical non-local effect. The solution of Equation (8) is
(8)$$w=A_{1}+A_{2}x+A_{3}\sin\left(\psi x\right)+A_{4}\cos\left(\psi x\right),$$
where $\psi =\sqrt{\frac{P_{F}+P_{\theta }-H}{EI+\bar{E}\bar{I}-\xi^{2}\left(P_{F}+P_{\theta }-H\right)}}$. Parameters $A_{i}\left(i=1,2,3,4\right)$ can be determined if both the mechanical boundary conditions and temperature distribution are given.

### 2.2. Temperature Field Governing Equation

For steady temperature distribution, the non-local heat conduction model can be written as follows [24]:(9)$$q+\gamma \frac{dq}{dx}+k\frac{d\theta }{dx}=0,$$
in which *q* is the heat flux, and κ is the heat conductivity. Parameter γ with the unit of meters (m) is the thermal non-local parameter. If we set γ=0 in Equation (10), the classical Fourier heat conduction model will be obtained.

Substituting the energy equation $\frac{dq}{dx}=Q$ into Equation (10), the non-local heat conduction model becomes
(10)$$Q+\gamma \frac{dQ}{dx}+k\frac{d^{2}\theta }{dx^{2}}=0.$$
where *Q* is the function of heat generation.

According to Ref. [19], the thermal boundary conditions and heat generation Q can be expressed as follows:(11)x=0:θ=0;x=L:q=0,
(12)$$Q=Q_{0}\sin\left(\frac{\pi }{2L}x\right).$$
where $Q_{0}=\frac{\pi^{5}I\kappa }{64AL^{4}\alpha_{t}}$.

Using Equations (11)–(13), temperature distribution in the micro/nano-beams can be obtained:(13)$$\theta \left(x\right)=\frac{Q_{0}}{\kappa }{\left(\frac{2L}{\pi }\right)}^{2}\left(\sin\left(\frac{\pi }{2L}x\right)+\frac{\gamma \pi }{2L}\cos\left(\frac{\pi }{2L}x\right)-\frac{\gamma \pi }{2L}\right).$$

## 3. Buckling Analysis

The buckling issue has a similar governing equation to the classical Euler–Bernoulli beam model. Therefore, there is no need to use an extra mechanical boundary condition. This ideology has been widely used to study the mechanical non-local effect of micro/nano-beams [31,32]. The boundary conditions of micro/nano-beams (c.f. Figure 2) can be expressed as follows:

Clamped–free micro/nano-beams (CF), i.e.,
(14)$$w=0,\;\frac{\partial w}{\partial x}=0,\;\text{at}\;x=0,$$
(15)M=0, V=0, at x=L.

Clamped–clamped micro/nano-beams (CC), i.e.,


(16)
$$w=0,\;\frac{\partial w}{\partial x}=0,\;\text{at}\;x=0,$$



(17)
$$w=0,\;\frac{\partial w}{\partial x}=0,\;\text{at}\;x=L.$$


Simply supported micro/nano-beams (SS), i.e.,


(18)
w=0, M=0, at x=0,



(19)
w=0, M=0, at x=L.


Simply supported–clamped micro/nano-beams (CS), i.e.,


(20)
w=0, M=0, at x=0,



(21)
$$w=0,\;\frac{\partial w}{\partial x}=0,\;\text{at}\;x=L.$$


Substituting Equation (9) into these boundary conditions, i.e., Equations (14)–(21), the critical load of micro/nano-beams with non-uniform temperature distribution can be obtained by solving the obtained equations, i.e.,
(22)$$P_{cr-\theta }=\frac{\left(EI+\bar{E}\bar{I}\right)\pi^{2}/{\left(\eta L\right)}^{2}}{1+{\left(\pi /\eta \right)}^{2}{\left(\xi /L\right)}^{2}}-P_{\theta }+H,$$
where η is the coefficient of equivalent length with constants 0.5, 0.7, 1 and 2 for boundary conditions CC, CS, SS and CF, respectively. Note that the external axial load *P* is written as $P_{cr-\theta }$ as it equals the critical load. Equation (22) indicates that the thermal effect and mechanical non-local effect reduce the critical load. However, the surface effect increases the critical load. In the absence of the thermal effect, Equation (22) can be reduced to
(23)$$P_{cr-\theta }=\frac{\left(EI+\bar{E}\bar{I}\right)\pi^{2}/{\left(\eta L\right)}^{2}}{1+{\left(\pi /\eta \right)}^{2}{\left(\xi /L\right)}^{2}}+H,$$

If both the thermal effect and small scale effect are neglected, Equation (22) becomes
(24)$$P_{0}=\frac{\pi^{2}EI}{{\left(\eta L\right)}^{2}}.$$

This simplification above is the same as the critical load of classical stability theory [33], which can be applied to prove the accuracy of the theoretical deduction.

## 4. Numerical Results and Discussion

Buckling of thermoelastic micro/nano-beams made of silicon is discussed in this section. The physical constants are [34] E=169 GPa, $\rho =2330\;kg/m^{3}$, $c_{E}=713\;J/\left(kgK\right)$, $\alpha_{t}=2.59\times 10^{-6}\;K^{-1}$, μ=0.22, $\kappa =156\;W/\left(mK\right)$ and $T_{0}=293\;K$. If there is no additional specifications, the following parameters are used: a=2h, L/h=30, $E_{s}=El$, ξ=2 nm, *l* = 2 nm, γ = 2 nm and τ=1 N/m. Note that *l* is the intrinsic material length, and it will be replaced with the surface layer thickness in the calculation [35]. Additionally, in the following description, Fourier heat conduction is abbreviated as the “Fourier model”, and the non-local heat conduction model is abbreviated as the “non-local model”, for simplicity.

### 4.1. Influence of Mechanical Small Scale Effect on Critical Load

Figure 3a–d are given to study the influence of the mechanical small scale effect on the critical load of buckling micro/nano-beams with various boundary conditions, i.e., CC with η=0.5, CS with η=0.7, SS with η=1 and CF with η=2. In the analysis process, parameter $P_{0}$ denotes the critical load predicted with the classical buckling model, and the parameter $P_{cr}$ is the critical load considering the mechanical small scale effect. The influence of the thermal effect on the critical load is neglected in Figure 3a–d. It can be seen that the critical load increases at the nano-scale as the mechanical small scale effect is considered. Take Figure 3d for example, compared to the critical load $P_{0}$ predicted with the classical stability theory, both the residual stress τ and the surface elastic modulus $E_{s}$ will lead to a larger critical load, $P_{cr}$, as the beam thickness is less than approximately 200 nm. The size-dependent critical load can also be observed in Figure 3a–c. Note that although the mechanical non-local effect decreases the critical load, the critical load still increases with the continuously decreased beam thickness. This is to say, the surface effect makes nano-beams “hard”. However, the mechanical non-local effect makes nano-beams “soft”. The influence of the mechanical small scale effect on the critical load of thermoelastic micro/nano-beams is clear.

### 4.2. Influence of Thermal Small Scale Effect on Critical Load

Figure 4 is used to describe the influence of the thermal non-local effect on the equivalent thermal force of thermoelastic micro/nano-beams with un-uniform temperature distribution. The parameter $P_{\mathrm{Non}-\mathrm{local}}$ denotes the equivalent thermal force predicted with the non-local heat conduction model, and the parameter $P_{\mathrm{Fourier}}$ denotes the equivalent thermal force predicted with the classical Fourier heat conduction model. Obviously, the thermal non-local effect reduces equivalent thermal force. Therefore, the thermal non-local effect increases the critical load of beams. This phenomenon can be explained with Equation (17), and which becomes more and more significant as the beam thickness is continuously diminished at the nano-scale. More important, as the beam thickness is larger than approximately 100 nm, the influence of the thermal non-local effect can be neglected and the classical Fourier heat conduction model is still useful. A similar phenomenon can also be seen in Figure 3a–d. In other words, both mechanical and thermal small scale effects can be neglected and the classical buckling model is still useful as the beam thickness is larger than approximately 100 nm. Otherwise, both of them must be considered in the thermoelastic buckling analysis. As a result, *h* = 100 nm is the critical thickness, which is determined with the small scale effect.

### 4.3. Influence of Combined Mechanical and Thermal Small Scale Effects on Critical Load

Figure 5a–d show the influence of the combined mechanical and thermal small scale effects on the critical load of buckling micro/nano-beams. The parameter $P_{\mathrm{cr}-\theta }$ denotes the critical load considering the thermal effect. In the influence region of critical thickness, it can be observed that the critical load increases when both a mechanical and thermal small scale effect are considered, and this size-dependent behavior depends on the mechanical boundary conditions. For example, as shown in Figure 5a–d, for a given beam thickness, the critical load increases with an increased equivalent length. Moreover, as mentioned before, the surface effect increases the critical load, mechanical non-local effect reduces the critical load and thermal non-local effect leads to a larger critical load. Due to the surface effect having a stronger influence than mechanical and thermal non-local effects on the critical load, the critical load finally increases at the nano-scale. This conclusion is different from the numerical result of Yu et al. [19], in which only the mechanical and thermal non-local effects are considered.

### 4.4. Influence of Other Key Factors on Critical Load

Figure 6 represents the influence of Poisson’s ratio on the critical load of thermoelastic micro/nano-beams. Both a mechanical and thermal small scale effect are considered in the analysis. Figure 6 indicates that at the macroscale, a larger critical load will be obtained if the Poisson effect is neglected. Although, the influence of the Poisson effect on the critical load begins to grow weaker with the decreased beam thickness at the nano-scale, and which can even be neglected if beam thickness is less than 10 nm. However, it should not be neglected in the region of 10 to 100 nm. It is thus clear that the Poisson effect was neglected in the buckling analysis of thermoelastic nano-beams [11,19], which may not be exact.

Figure 7a–d show the influence of the thickness-to-length ratio on the critical load of thermoelastic micro/nano-beams. Obviously, for a given beam thickness, the influence of the small scale effect on the critical load becomes more significant as the thickness-to-length ratio increases. In addition, it can be seen from Figure 7a–d that the critical load increases with an increased equivalent length if beam thickness is given. This phenomenon can also be found in Figure 5a–d, and which can be used to verify the accuracy of the numerical simulation.

## 5. Conclusions Remarks

In this paper, thermoelastic buckling of micro/nano-beams with non-uniform temperature distribution is discussed based on the non-local elastic theory and surface elastic theory. Non-uniform temperature distribution in the micro/nano-beams is first calculated based on the non-local heat conduction model. The temperature-induced equivalent thermal load is regarded as the component of the external axial load. The analytical solution of the critical load is obtained. The influence of the surface effect and both mechanical and thermal non-local effects on thermoelastic buckling behaviors of micro/nano-beams is discussed in detail. The numerical simulation indicates that

A critical thickness is proposed: as the beam thickness is less than the critical thickness, the influence of both mechanical and thermal small scale effects on the buckling load must be considered. As beam thickness is larger than the critical thickness, the influence of both mechanical and thermal small scale effects on the buckling load can be neglected, and the classical buckling model can be applied for theoretical prediction of the buckling load. In short, the critical thickness can be used to distinguish the influence region of the small scale effect in practical engineering applications.The critical load depends on the boundary conditions, the length-to-thickness ratio and Poisson’s ratio. These findings are useful to the design of micro/nano-beams.In the influence region of the small scale effect, the combined small scale effects can give rise to a larger critical load. Specifically, the surface effect leads to a “hard beam”, mechanical non-local effect leads to a “soft beam” and thermal non-local effect leads to a larger critical load. The surface effect has great influence on the critical load compared to both mechanical and thermal small scale effects.

To summarize, these findings above indicate that the critical load of micro/nano-beams depends not only on the structural size but also the thermoelastic coupling. Hopefully, these conclusions are useful to a buckling analysis, especially for thermoelastic micro/nano-beams subjected to non-uniform temperature distribution. In the future, an experimental analysis will be carried out to validate the proposed model.

## Figures and Tables

**Figure 1 materials-16-06390-f001:**
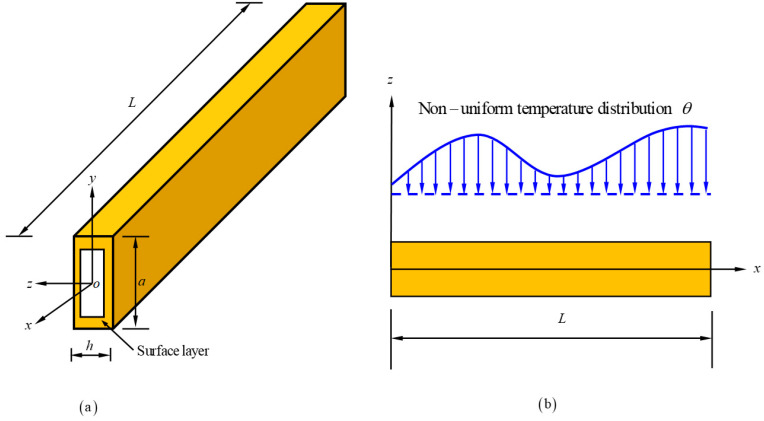
Configuration of size-dependent micro/nano–beams with non-uniform temperature distribution. (**a**) Structural diagram of the beam (**b**) Schematic diagram of the force acting on a beam under non-uniform temperature.

**Figure 2 materials-16-06390-f002:**
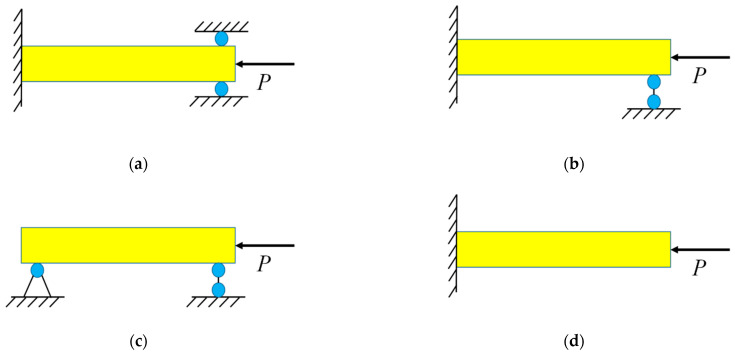
Mechanical boundary conditions of thermoelastic micro/nano–beams: (**a**) CC, (**b**) CS, (**c**) SS and (**d**) CF.

**Figure 3 materials-16-06390-f003:**
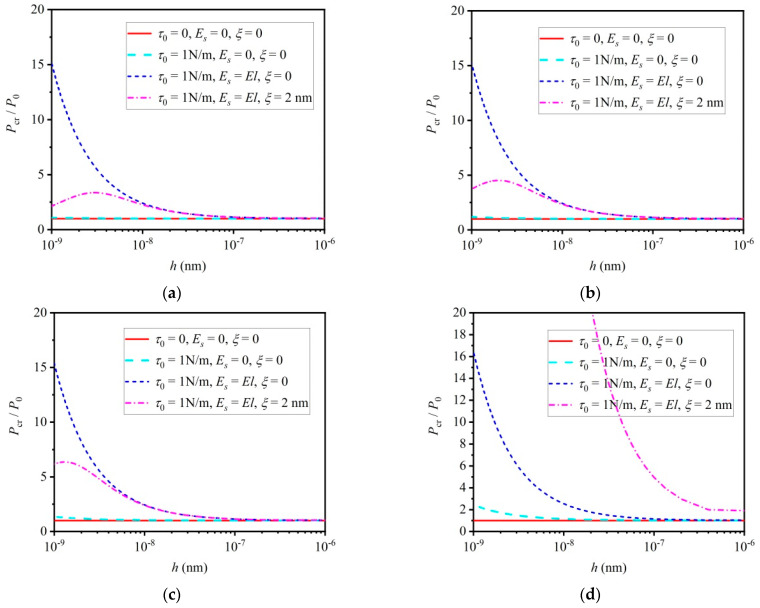
Influence of mechanical small scale effect on critical load of buckling micro/nano–beams without considering thermal effect: (**a**) CC with η=0.5, (**b**) CS with η=0.7, (**c**) SS with η=1 and (**d**) CF with η=2.

**Figure 4 materials-16-06390-f004:**
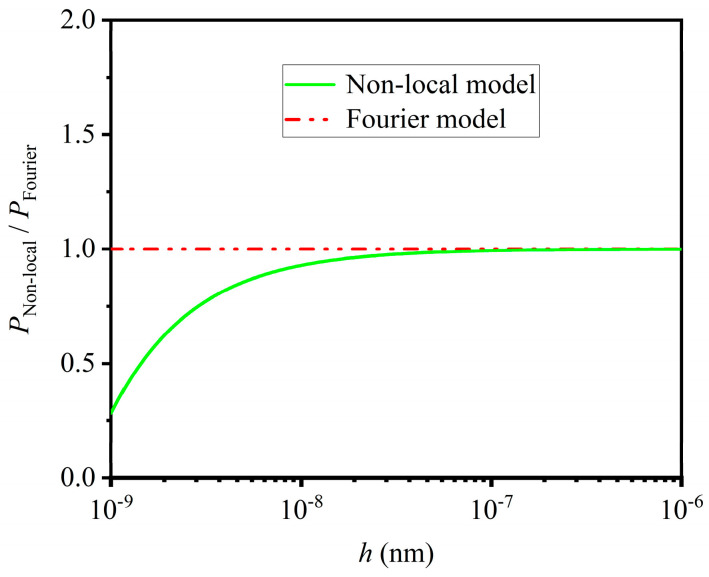
Influence of thermal non–local effect on equivalent thermal force of thermoelastic micro/nano–beams.

**Figure 5 materials-16-06390-f005:**
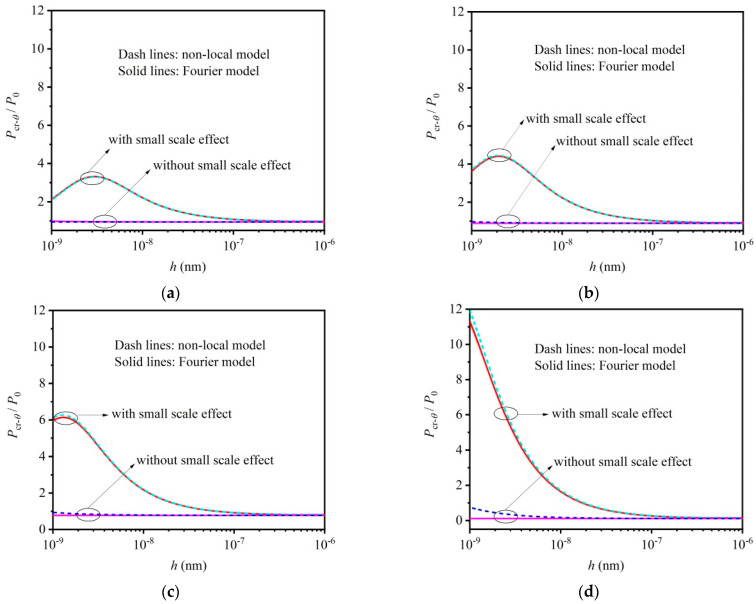
Influence of combined mechanical and thermal small scale effects on critical load of thermoelastic micro/nano–beams: (**a**) CC with η=0.5, (**b**) CS with η=0.7, (**c**) SS with η=1 and (**d**) CF with η=2.

**Figure 6 materials-16-06390-f006:**
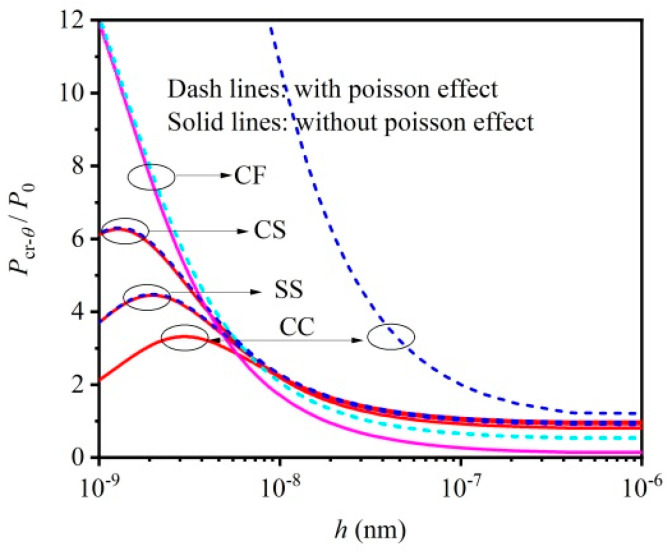
Influence of Poisson’s ratio on critical load of thermoelastic micro/nano–beams considering mechanical and thermal small scale effect: CC with η=0.5, CS with η=0.7, SS with η=1 and CF with η=2.

**Figure 7 materials-16-06390-f007:**
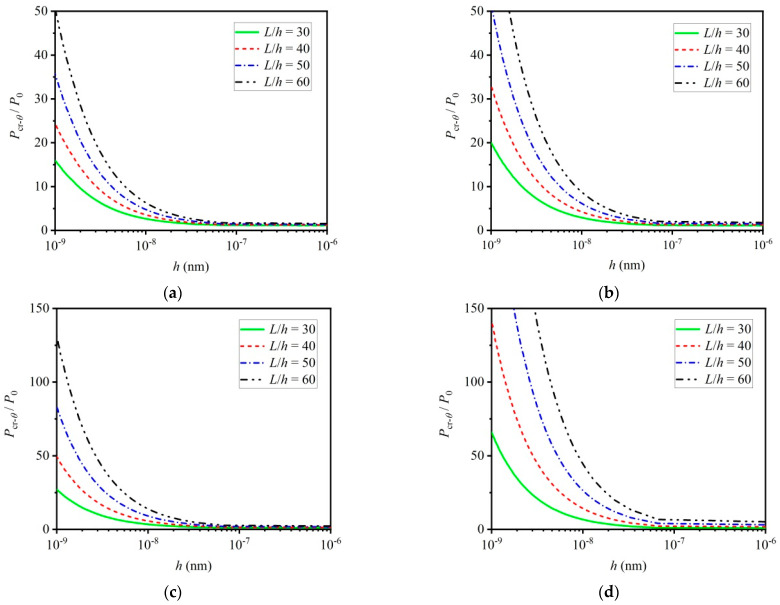
Influence of thickness-to-length ratio on critical load of thermoelastic micro/nano–beams considering mechanical and thermal small scale effect: (**a**) CC with η=0.5, (**b**) CS with η=0.7, (**c**) SS with η=1 and (**d**) CF with η=2.

## Data Availability

Not applicable.

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
