# Peer review of "A Buckling Analysis of Thermoelastic Micro/Nano-Beams Considering the Size-Dependent Effect and Non-Uniform Temperature Distribution"

_materials, 2023, doi:10.3390/ma16196390_

Round 1

Reviewer 1 Report

The manuscript analyses the Buckling analysis of thermoelastic micro/nano-beams considering the size-dependent effect and non-uniform temperature distribution. The mechanical governing equation is derived considering both the surface effect and the mechanical non-local effect. The non-local heat conduction model is first used to predict temperature distribution in micro/nano-beams. The manuscript contains some merits, and the content is interesting. However, some major problems must be carefully addressed and explained for reconsideration:

1.         The presentation structure in the introduction to lead the research gap and discuss the new contribution is somehow unclear and still messed up. The authors are hence advised to revise the introduction section to present deficiencies or shortcomings of other studies to make a bridge or research gap to introduce the novelty of their work instead of a long introduction paragraph.

2.         The formulations are presented without introducing a suitable reference for each one. Authors should provide appropriate references for each basic formula.

3.         Authors need to present the boundary conditions representation as figures also.

4.         Authors should incorporate the verification study of the present model with the existing available literature study and at least present two verification examples.

5.         Explain all the figures in section 4 in detail with reason so that readers can understand the behaviour of the microbeams.

6.         The reviewer suggests revising the Conclusion. It is recommended to discuss the findings of the research by emphasizing engineering applications.

7.         The authors should also include one or two sentences to illustrate the Future recommendations" after the conclusions to discuss the prospects and recommendations.

8.         I do not find any of the latest research papers in the reference. All the cited articles are from 2020 or before. Why?

9.         Why did the authors not consider the porosity or voids effect in the micro beams buckling behaviour?

10.       Why did the authors consider only non-uniform temperature distributions? Why not other temperature distributions like uniform, linear or sinusoidal type?

Reviewer 2 Report

The work is devoted to an urgent topic: the loss of stability of thermoelastic micro/nanobanks depending on size-dependent parameters. Different theories that are used to describe the heat equation are compared. The research is interesting and suitable for publication in the journal. However, it is necessary to eliminate some shortcomings:

1. The literature analysis is incomplete. In recent years, a huge amount of literature has been devoted to this topic, but in the list of references in this article there are only five sources for the last three years. It is necessary to refer to the latest articles on this topic. For example, J Awrejcewicz, AV Krysko, A Smirnov, LA Kalutsky, MV Zhigalov..., Mathematical modeling and methods of analysis of generalized functionally gradient porous nanobeams and nanoplates subjected to temperature field //Meccanica 57 (7), 1591-1616; AV Krysko, VA Krysko, IV Papkova, TV Yakovleva General Theory of Porous Functionally Gradient MEMS/NEMS Beam Resonators Subjected to Temperature Field 2022/5/30 2022 29th Saint Petersburg International Conference on Integrated Navigation Systems (ICINS)

2. It is necessary to compare the results obtained with the results of other authors. These may be other theories to account for the size-dependent behavior of the system.

3. The Euler-Bernoulli hypothesis is used, which is designed for thin beams (30 < L/h <100). And in the numerical experiment, the ratio of length to thickness L/h = 5. These are thick beams that should be described by a higher-order theory (Timoshenko, a third-order hypothesis).

4. Remove the big gap on page 5

5. Describe in more detail the solving methods of the equations

Round 2

Reviewer 1 Report

The revised manuscript is accepted for publication.

Reviewer 2 Report

Dear authors!

Higher resolution drawings should be made.